# Hepadnavirus Infection in a Cat with Chronic Liver Disease: A Multi-Disciplinary Diagnostic Approach

**DOI:** 10.3390/vetsci10120668

**Published:** 2023-11-24

**Authors:** Paolo Capozza, Francesco Pellegrini, Michele Camero, Georgia Diakoudi, Ahmed Hassan Omar, Anna Salvaggiulo, Nicola Decaro, Gabriella Elia, Leonardo Catucci, Barbara Di Martino, Paola Fruci, Letizia Tomassini, Elvio Lepri, Vito Martella, Gianvito Lanave

**Affiliations:** 1Department of Veterinary Medicine, University of Bari, 70010 Bari, Italy; paolo.capozza@uniba.it (P.C.); francesco.pellegrini@uniba.it (F.P.); michele.camero@uniba.it (M.C.); georgia.diakoudi@uniba.it (G.D.); ahmed.omar@uniba.it (A.H.O.); anna.salvaggiulo@uniba.it (A.S.); nicola.decaro@uniba.it (N.D.); gabriella.elia@uniba.it (G.E.); gianvito.lanave@uniba.it (G.L.); 2ACV Triggiano S.r.l., 70019 Triggiano, Italy; leocatucc@libero.it; 3Faculty of Veterinary Medicine, University of Teramo, 64100 Teramo, Italy; bdimartino@unite.it (B.D.M.); pfruci@unite.it (P.F.); 4Istituto Zooprofilattico Sperimentale dell’Umbria e delle Marche, 06126 Perugia, Italy; l.tomassini@izsum.it; 5Department of Veterinary Medicine, University of Perugia, 06126 Perugia, Italy; elvio.lepri@unipg.it

**Keywords:** cat, domestic cat hepadnavirus, serum, liver, hepatitis

## Abstract

**Simple Summary:**

Domestic cat hepadnavirus (DCH) is a newly identified virus, similar to the hepatitis B virus (HBV). The pathogenicity of DCH in its host needs further investigation. In this report, a case of DCH infection was described in a cat with clinical signs and serum chemistry markers indicative of chronic hepatic disease using a multi-disciplinary diagnostic approach, combining clinical, pathological, virological, and serological information, mirroring the diagnostic approach used for HBV in human patients. Improving the intra-vitam diagnostics for DCH will be useful in developing specific treatment and control strategies.

**Abstract:**

A 3-year-old female stray, shorthair cat, with clinical signs and serum chemistry markers indicative of hepatic disease, was diagnosed with domestic cat hepadnavirus (DCH) infection. Coupling molecular and serological data, the infection was seemingly contextualized into a chronic phase, since IgM anti-core antibodies, a marker of early-stage Hepatitis B Virus (HBV) infection, were not detected. However, the cat possessed IgG anti-core, a common indicator of chronic HBV infection in human patients and did not show seroconversion to the anti-DCH surface antigen, considered protective during HBV infection and associated with long-term protective immunity. On genome sequencing, the DCH strain showed 98.3% nucleotide identity to strains previously identified in Italy.

## 1. Introduction

Domestic cat hepadnavirus (DCH), discovered in 2018, is genetically similar to human hepatitis B virus (HBV). The DCH genome consists of approximately 3200 bases of circular, partially double-stranded DNA [1]. In molecular investigations from ccountries in Asia, Australia, the USA, and Europe, DCH has been identified in 0.2 to 18% of cats [2,3,4,5,6,7]. This is likely an underrated estimation of the prevalence of DCH, since anti-DCH core antibodies (anti-DCHc) have been revealed in about 25% of the tested animals [8]. The pathobiology of this novel hepadnavirus is still largely unknown, but a possible etiopathogenetic role has been hypothesized in the development of feline liver disease. Hepatic histological alterations suggestive of inflammation and neoplasia, as documented for HBV, have been described in DCH-positive cats with chronic hepatitis or hepatocellular carcinoma (HCC) [9]. Furthermore, a positive correlation has been observed between the hepatic markers and the presence of DCH DNA [2,3,4,6,10,11,12,13,14,15,16,17]. More recently, DCH DNA was detected in serum and in inflammatory and non-inflammatory peritoneal effusion samples but not in liver biopsy from cats infected with DCH with hematological and biochemical alterations [5]. In addition, viremia from DCH has also been identified in cats with normal or mildly elevated liver enzymes [18]. Coinfection with feline retroviruses and age > 2 years are additional risk factors for DCH viremia [3,6,8,10,17]. The disease spectrum and the course of HBV infection vary in terms of severity, progression, and prognosis, ranging from acute inapparent infections to progressive chronic hepatitis [19]. Antigens, antibodies, and viral DNA may be determined and combined with the haematological parameters to identify HBV and predict the stage of infection in human patients [19,20]. Accordingly, DCH diagnosis could benefit from a combined virological/serological approach, following the outline of the HBV diagnostic algorithm in human patients. We, herein, describe a case of DCH infection in a cat with evidence of hepatic illness, for whom intra-vitam diagnosis was obtained with direct and indirect diagnostics. The full-genome sequence of the virus infecting the cat was also generated.

## 2. Case Description

A 3-year-old female stray, shorthair (DSH) cat was referred to a private veterinary clinic in March 2021 for lethargy, anorexia, and weakness. On physical examination, the cat showed a temperature in the physiological range (38 °C), pale mucous membranes, jaundice, and weight loss. After the initial physical examination, a baseline assessment was carried out, consisting of a complete blood count, serum biochemistry, urinalysis, and retroviral screening, using a point-of-care (POC) immunochromatographic test (SNAP FIV/FeLV Combo Test, IDEXX). Abnormalities in serum biochemistry included elevations of alkaline phosphatase (ALP; 160 International Unit (IU)/L; Reference Interval (RI) 14–62), gamma-glutamyl transferase (GGT; 6.2 IU/L; RI 0.0–1.0), total bilirubin (4.42 mg/dL; RI 0.14–0.25), globulins (5.5 g/dL, RI 2.9–4.3), γ-globulin (33.6%; RI 15.0–28.0), and amylase (1755 IU/L; RI 648–1262). Cholesterol (91 mg/dL; RI 112–194), total iron (24 μg/dL; RI 55–152), and saturation (6.8%; RI 20–53) were decreased. A complete blood count revealed leucocytosis (increased white blood cells (WBCs) 46.7 × 10^3^/μL, 5.5–12 × 10^3^/μL) associated with anaemia (3.51 × 10^6^/μL; RI 6.0–9.5 × 10^6^/μL). Using the POC assay, the cat tested negative for retroviral infections. Haematological parameters were monitored repeatedly over four additional time points (April 2021, October 2021, November 2021, and February 2022), evidencing a persistent elevation in ALP, GGT, total bilirubin, and WBC values, along with anaemia and leucocytosis. Aspartate aminotransferase (AST) and alanine aminotransferase (ALT) values were out of range at two and three time monitoring points, respectively (Table 1). 

Abdominal ultrasonographic examinations performed in March 2021 and repeated in April and November 2021 (Figure 1) showed chronic liver disease and chronic gall-bladder abnormalities, with mild cholestasis. 

A liver biopsy specimen (collection date November 2021) was fixed in 10% buffered formalin, processed for paraffin sectioning, and cell staining was performed using haematoxylin and eosin. On histological examination, almost all the portal tracts were characterized by moderate fibrosis with mild lymphoplasmacytic infiltrate and ductular hyperplasia, leading to a diagnosis of lymphoplasmacytic diffuse mild chronic periportal hepatitis (Figure 2A). Bacterial or fungal elements were not evident. Upon virological investigations performed on the liver biopsy sample and on two serum specimens collected in November 2021 and February 2022, DCH DNA was identified using quantitative PCR (qPCR) [10] only from the liver sample, with a viral load of 2.9 × 10^1^ DNA copies/mL of extracted DNA template. 

When retesting all the three specimens with a qualitative PCR protocol [1], both liver and serum samples, collected in November 2021, were positive, whilst the serum collected in February 2022, 3 months after the initial diagnosis, tested negative for DCH. All the specimens were negative for feline calicivirus, feline herpesvirus type 1, feline panleukopenia parvovirus, feline coronavirus, and feline retroviruses, thus ruling out co-infections [21,22,23,24,25,26]. The immunohistochemistry (IHC) assay, performed on the liver biopsy by using a rabbit polyclonal anti-HBV core antigen (HBcAg) (Thermo Fisher Scientific, Waltham, MA, USA), as previously described [11,12], revealed cytoplasmic immunoreactivity with focal strong intensity (Figure 2B), confirming the presence of DCHcAg. 

To characterize the DCH strain, the viral DNA detected in the liver sample was enriched using a rolling circle amplification (RCA) with a TempliPhi and RCA primer mix. Subsequently, a total of 1 μg of a dilution 1:100 of the purified RCA product was used as a template for a library prepared with the Ligation sequencing kit 1D SQK-LSK110 (ONT^TM^, Oxford, UK) for MinION sequencing (Oxford Nanopore Technology^TM^). The genome was assembled using Geneious Prime version 2021.2 (Biomatters Ltd., Auckland, New Zealand). The DCH sequence was aligned with other DCH sequences selected from the GenBank (NCBI) database (examined in May 2023) using the Geneious Alignment plugin. The substitution model parameters for phylogenetic inference and assessment of selection pressure were selected using MEGA X version 10.0.5 [27]. The phylogenetic reconstruction was carried out using the Maximum Likelihood method and Tamura–Nei model (four parameters) [28], a discrete gamma distribution, and invariable sites (six categories), with bootstrapping over 1000 replicates. 

In order to investigate the serological response against DCH, the sera samples collected in November 2021 and in February 2022 were tested using two in-house ELISA assays based on a recombinant DCHcAg and DCH surface antigen (DCHsAg) [8,29]. Only IgG anti-DCHcAg was detected, whilst both sera resulted as negative for IgM anti-DCHcAg and IgG anti-DCHsAg. Sera obtained before November 2021 were not available for virological and serological examinations. The cat was found dead without traumatic injuries in May 2022, and post-mortem examination could not be made.

## 3. Discussion and Conclusions

Since the identification of DCH in cats in 2018, numerous global studies have been conducted to assess this virus’s epidemiology and potential role in feline liver disease [1,2,3,4,8,9,10,11,12,13,14,15,16,18]. An association has been identified between DCH infection and clinical signs or markers related to hepatic damage [2,3,4,6,10,11,12,13,14,15,16,17]. DCH-infected cats are likely to have elevated ALT levels [10,15]. Cats with haematochemical profiles indicative of hepatic damage are about three-times more likely to test positive for DCH than animals with normal haematochemical parameters [15]. These findings suggest the potential contribution of DCH in the development of hepatic disease in these animals. Based on the clinical signs, history, instrumental diagnostic analysis, laboratory results, and viral investigation, the case described in this report was framed as a chronic periportal hepatitis associated with DCH infection. Lymphocytic cholangio-hepatitis was suspected due to the increase in the serum concentration in ALT, ALP, GGT, bile acids, and total bilirubin, associated with the alteration to the leukogram [30]. Subsequently, the histological examination performed on liver biopsy revealed moderate fibrosis with mild lymphoplasmacytic infiltrate and ductular hyperplasia in almost all portal areas, a picture compatible with periportal hepatitis [31,32]. 

In a retrospective investigation, out of 71 cat liver biopsies, a high prevalence of DCH was described in cats with chronic hepatitis (43%) and hepatocellular carcinoma (28%), compared with liver tissues without histological alterations [9]. Animals with chronic hepatitis positive for DCH in PCR showed lymphocytic periportal inflammation, with some hepatic areas presenting inflammation at the portal–lobular interface. Additionally, lymphocytes and plasma cells were dispersed within sinusoidal spaces where they occasionally clustered. Neutrophils were uncommon and, when present, were close to single-cell regions or to necrotic areas. Fibrosis irregularly spanned the portal regions and affected regional sinusoids [9]. In a more recent investigation, Thai researchers established a correlation between DCH and liver histological parenchymal disease (HPD). Specifically, they revealed a higher proportion of DCH-positive HPD cases, with portal areas surrounded by thick bands of fibrosis connecting the portal structures in some tracts and showing infiltrates of numerous lymphocytes and plasma cells [11,12].

In our analysis, IHC staining revealed cytoplasmic reactivity for DCH that appeared as multifocal and diffuse staining in the hepatocytes and bile duct epithelial cells adjacent to the areas of inflammation and fibrosis. These findings are comparable to the results of a previous Thai investigation [12], in which the presence of DCH was confirmed via IHC performed using a polyclonal antibody to the HbcAg antigen, with a positive signal being observed in the hepatocytes and biliary epithelial cells. The IHC assay against both HBsAg and HBcAg is well described in human medicine, and this pattern is correlated with viral replication activity and virus titre. In feline species, a correlation between the immunohistochemical pattern and the above-mentioned features is not yet possible. Therefore, the results obtained should be interpreted considering the margin of variability that has remained untested. 

Interestingly, DCH DNA was detected in the serum sample and in liver biopsy, collected in November 2021, using a PCR with primers specific for DCH but not in a follow-up serum sample collected in February 2022. During HBV infection, some patients may exhibit a status of chronic inactive carrier or a form of HBeAg-negative chronic hepatitis, related to mutations in the HBV genome, with the hepadnavirus DNA being detectable in a negligible titre or being undetectable in the blood [19,33]. The cat possessed IgG anti-DCHcAg, a common indicator of chronic infection in HBV-infected human patients, and did not show seroconversion to anti-DCHsAg, considered protective during HBV infection and associated with the recovery from infection [19,33]. In addition, both sera were negative for IgM anti-core, a serological marker indicative of a very early stage of HBV infection. Accordingly, coupling molecular and serological data, the DCH infection was seemingly contextualized into a chronic phase.

The full-length genome sequence of 3185 bp was generated for the DCH strain (GenBank accession nr. OR389995) (Figure 3). 

In genome-wise sequence analysis, the DCH strain showed nucleotide identity between 97 and 98% with other Italian strains [8,10,18]. On phylogenetic analysis (Figure 3) using a whole-genome sequence, the DCH strain ITA/2021/665 was segregated with Malaysian, Thai, Australian, European, and American DCH viruses in genotype A. Strain ITA/2021/665 fell into a well-defined sub-clade, proposed as clade A2, along with Italian, Russian, Turkish, and American strains (Figure 3), distinguishable from sub-clade A1 that contained DCH viruses from Malaysia, Thailand, and Australia, and from the Japanese DCH strain Rara, genotype. 

In conclusion, this study provides information on the clinical case of a cat with clinical signs and serum chemistry markers indicative of hepatic disease related to DCH chronic infection. This hypothesis was supported by abdominal ultrasonographic examinations, histological and immuno-histological analysis of liver biopsy, the persistent alteration of the markers of hepatocellular functionality, and the positivity to IgG anti-DCHcAg. Gathering information on DCH is pivotal to understanding its pathobiological role and developing specific treatment and control strategies. Since chronic HBV infection in human patients can evolve across different stages with marked fluctuations in virus activity and even with virus-undetectable phases, combining clinical, bio-chemical, virological, and serological data must be considered as an elective approach for DCH diagnosis. 

## Figures and Tables

**Figure 1 vetsci-10-00668-f001:**
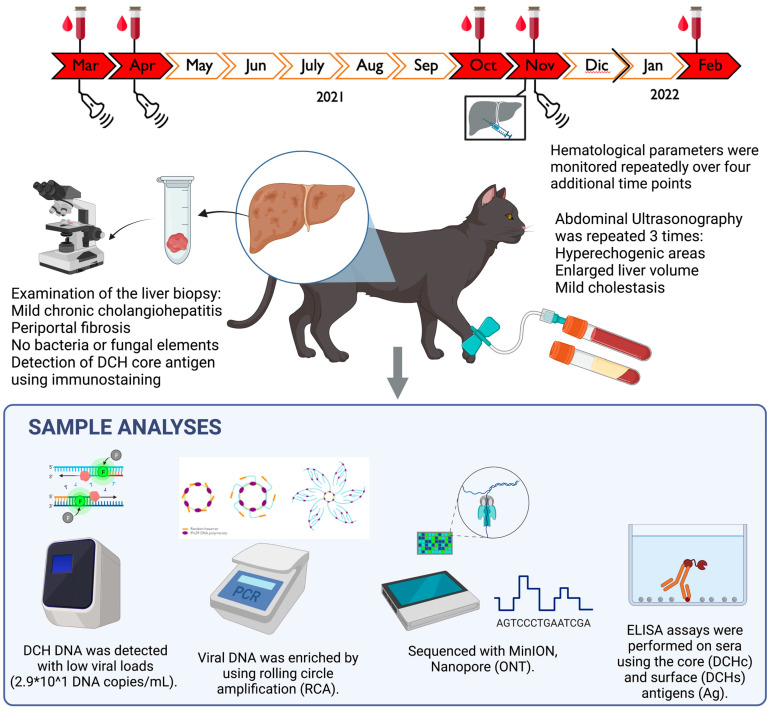
Phases of the investigation.

**Figure 2 vetsci-10-00668-f002:**
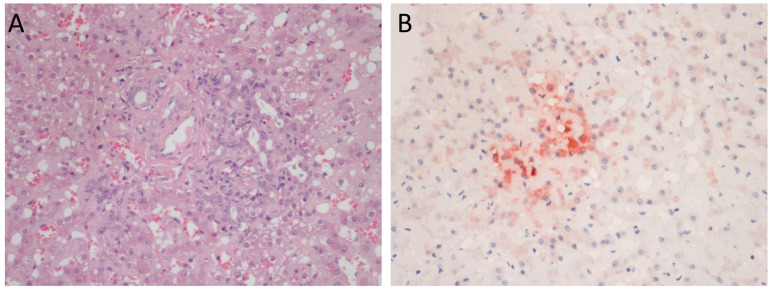
(**A**) Lymphoplasmacytic mild chronic periportal hepatitis with ductular hyperplasia. H&E, 400× magnification; (**B**) multifocal cytoplasmic immunostaining. Domestic cat hepadnavirus immunohistochemistry, 400× magnification.

**Figure 3 vetsci-10-00668-f003:**
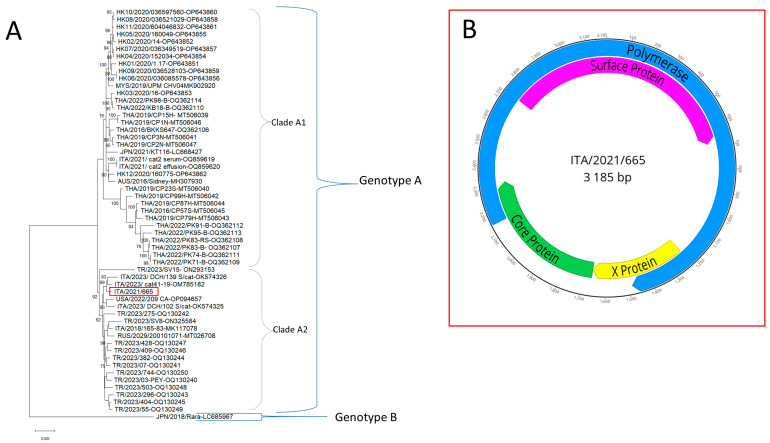
(**A**) Phylogenetic reconstruction based on the complete genomes of domestic cat hepadnavirus obtained in this study (red square) and sequences obtained from the databases. Statistical support was determined using 1000 bootstrap replicates. Bootstrap values greater than 75% are shown. Genotypes A and B, and clades A1 and A2 within genotype A are indicated. The scale bar represents the number of nucleotide substitutions per site. (**B**) Structure of the DCH genome.

**Table 1 vetsci-10-00668-t001:** Blood parameters of the patient. Parameters that exceed the normal ranges are in bold.

Sampling Date	RBC 6.0–9.5 × 10^6^/μL	WBC 5.5–12 × 10^3^/μL	PLT 130–400 × 10^3^/μL	CPK 52–295 IU/L	AST 16–46 IU/L	ALT 33–70 IU/L	ALP 14–62 IU/L	GGT 0–1 IU/L	Total Bilirubin 0.14–0.25 mg/dL	DCH DNA	Notes
18 March 2021	**3.51**	**46.7**	382	146	26	30	**160**	**6.2**	**4.42**	NT	
9 April 2021	**5.11**	**60.5**	397	62	30	**101**	**94**	**5.0**	**4.16**	NT	
14 October 2021	**6.33**	**27.7**	**482**	68	**167**	**162**	**63**	**2.8**	**1.87**	NT	
25 November 2021	**5.78**	**23.8**	256	76	20	38	**76**	**4.0**	**1.49**	Pos.	Liver biopsy positive for hepadnavirus (2.9 × 10^1^ DNA copies/mL)
11 February 2022	**3.68**	**25.7**	348	148	**183**	**1285**	**93**	**4.1**	**0.31**	Neg.	

Abbreviations: RBC, red blood cells; WBC, white blood cells; PLT, platelets; CPK, creatine phosphokinase; AST, aspartate transaminase; ALT, alanine transaminase; ALP, alkaline phosphatase; GGT, gamma glutamyl transferase; IU, International Unit; L, Liter; mg, milligrams; μL, microlitres; mL, millilitres; DCH, domestic cat hepadnavirus, Neg., negative; Pos., positive; NT, not tested.

## Data Availability

The data that support the findings of this study are available from the corresponding author upon reasonable request.

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
