# Peer review of "Hepadnavirus Infection in a Cat with Chronic Liver Disease: A Multi-Disciplinary Diagnostic Approach"

_vetsci, 2023, doi:10.3390/vetsci10120668_

Round 1

Reviewer 1 Report

Comments and Suggestions for Authors

Review of Hepadnavirus infection in a cat with chronic liver disease by Capozza et al.  Where was this cat living during this study?  Was the cat fed a diet to help with the liver disease?  What was the outcome of the cat?  Since DCH has been reported in cats since 2018, what makes this case unique?  This paper mentions DCH is closely related to human hepatitis virus.  Is there a suggestion that it is zoonotic or reverse zoonotic?

Sample Analyses insert.  Was the analyzed sample blood or the biopsy?

Sentence beginning on line 182, and uncited subsequent sentences- please provide a reference.

Author Response

Comments from Reviewer 1 (R1)

R1.1: Review of Hepadnavirus infection in a cat with chronic liver disease by Capozza et al. Where was this cat living during this study?  

Reply to R1.1: The animal was a stray cat with outdoor lifestyle. It lived in the urban area, and a caretaker provided the animal with food and water.

R1.2: Was the cat fed a diet to help with the liver disease?  What was the outcome of the cat?  

Reply to R1.2: The cat was given an adequate diet to alleviate the liver disease. The animal continued to exhibit the biochemical parameters typical of chronic liver failure.

R1.3: Since DCH has been reported in cats since 2018, what makes this case unique?  

Reply to R1.3: Domestic cat hepadnavirus (DCH, family Hepadnaviridae) was first reported from whole blood samples of domestic cats in Australia in 2018, and from cat serum samples in Italy in 2019. It was reported in cats with viraemia (6.5–10.8%), chronic hepatitis (43%) and hepatocellular carcinoma (28%) usually by testing collections of samples from cohorts of animals. Recent studies also suggest that DCH resembles the human hepatitis B virus (HBV) in terms of hepatic tropism and pathobiology. Diagnosis of human hepatitis B infection/disease is complex and requires a multi-disciplinal approach, based on the clinical picture, serology, and virology, eventually coupled with biopsy sampling. Our case was rather unique since we were able to make a diagnosis of DCH based on the clinical history, on bioptic analysis, specific immune histology, and serology and molecular investigations. This case report underlines the importance of an interdisciplinary approach in formulating a diagnosis of DCH, similar to what happens in the diagnostics of HBV hepatitis in human patients.

R1.4: This paper mentions DCH is closely related to human hepatitis virus.  Is there a suggestion that it is zoonotic or reverse zoonotic?

Reply to R1.4: The observation of the referee is quite interesting, Hepatitis B virus (HBV) (the prototype of the family Hepadnaviridae), is a human pathogen and it does not infect cats or other animals. However, oddly, HBV-like virus has been identified in pigs in a Brazilian study, although this has not been confirmed elsewhere (Vieira YR, dos Santos DR, Portilho MM, Velloso CE, Arissawa M, Villar LM, Pinto MA, de Paula VS. Hepadnavirus detected in bile and liver samples from domestic pigs of commercial abattoirs. BMC Microbiol. 2014 Dec 11; 14:315. doi: 10.1186/s12866-014-0315-2. PMID: 25495746; PMCID: PMC4269919). The HBV-derived preS1 peptide does not bind to pig NTCP. However, pig hepatocytes expressing the human NTCP support HBV replication (Lempp FA, Wiedtke E, Qu B, Roques P, Chemin I, Vondran FWR, Le Grand R, Grimm D, Urban S. Sodium taurocholate cotransporting polypeptide is the limiting host factor of hepatitis B virus infection in macaque and pig hepatocytes. Hepatology. 2017 Sep;66(3):703-716. doi: 10.1002/hep.29112. Epub 2017 Jul 18. PMID: 28195359).

DCH has been detected thus far only in cats and dogs. In 2022 an Italian study identified hepadnavirus DNA in 6.3% (40/635) of canine serum samples, although the viral load in positive sera was low. On genome sequencing, the canine hepadnaviruses revealed high nucleotide identity (about 98%) and similar organization to the domestic cat hepadnavirus, suggesting the possibility of a free circulation of hepadnaviruses among domestic carnivores, rather than different viral species with a specific host range (Diakoudi G, Capozza P, Lanave G, Pellegrini F, Di Martino B, Elia G, Decaro N, Camero M, Ghergo P, Stasi F, Cavalli A, Tempesta M, Barrs VR, Beatty J, Bányai K, Catella C, Lucente MS, Buonavoglia A, Fusco G, Martella V. A novel hepadnavirus in domestic dogs. Sci Rep. 2022 Feb 21;12(1):2864. doi: 10.1038/s41598-022-06842-z. PMID: 35190615; PMCID: PMC8860997).

In a more recent investigation, the entry receptor for DCH has been identified. The researchers demonstrated a significant similarity in cellular entry receptors between HBV and DCH. Both viruses utilize sodium/bile acid cotransporter (NTCP) as a functional receptor, and more importantly, the HBV-derived preS1 peptide can bind to cat NTCP. On the other hand, the DCH-derived preS1 peptide can bind to human NTCP. Furthermore, they demonstrated that the DCH-derived preS1 peptide binds to NTCPs derived from a broad range of animal species, suggesting that DCH has the potential for interspecies transmission (Shofa M, Ohkawa A, Kaneko Y, Saito A. Conserved use of the sodium/bile acid cotransporter (NTCP) as an entry receptor by hepatitis B virus and domestic cat hepadnavirus. Antiviral Res. 2023 Sep; 217:105695. doi: 10.1016/j.antiviral.2023.105695. Epub 2023 Aug 1. PMID: 37536428).

Overall, the potential of DCH to infect human host or other animal species is unclear and there is no evidence of a zoonotic potential, thus far. We did not mention this in the manuscript, since we did not think it could add something relevant for the main message of our report.

R1.5: Sample Analyses insert.  Was the analyzed sample blood or the biopsy?

Reply to R1.5: Blood and liver biopsy samples were tested with molecular tests to detect DCH DNA. ELISA assay was conducted on a sera sample to evaluate the presence of antibodies (IgM and IgG). Finally, histological and immuno-histological analyses were performed on liver biopsy.

R1.6: Sentence beginning on line 182, and uncited subsequent sentences- please provide a reference.

Reply to R1.6: We added the specific reference in the text.

Reviewer 2 Report

Comments and Suggestions for Authors

Dear Authors,

I revised the work entitled "Hepadnavirus infection in a cat with chronic liver disease: a multi-disciplinary diagnostic approach". I found the topic very interesting, extremely current and the case report presented in a truly linear and explanatory way. The study is well structured, full of details and diagnostic techniques, from serology, to molecular biology with phylogenetic tree, to the identification of viral antigens using IHC. In my opinion, the title is appropriate, the introduction focuses the topic well, the methods are adequately described and the discussions are detailed and supported by the results. The most important studies present in the literature have all been cited and the bibliography is updated. Diagrams, figures and photos are all high quality and self explainig. The presence of Hepadnavirus in cats is a recent finding dated back only to 2018, little is still known about the life cycle, pathogenesis, epidemiology, etc. of this virus, so in my opinion every single case is worthy of being taken into consideration, even more if complete like this. It's a pity that it wasn't possible to perform a necropsy after the subject's death also to collect other samples.

Comments on the Quality of English Language

Minor editing of English language required

Author Response

Comments from Reviewer 2 (R2)

Dear Authors, I revised the work entitled "Hepadnavirus infection in a cat with chronic liver disease: a multi-disciplinary diagnostic approach". I found the topic very interesting, extremely current and the case report presented in a truly linear and explanatory way. The study is well structured, full of details and diagnostic techniques, from serology to molecular biology with phylogenetic tree, to the identification of viral antigens using IHC. In my opinion, the title is appropriate, the introduction focuses the topic well, the methods are adequately described, and the discussions are detailed and supported by the results. The most important studies present in the literature have all been cited and the bibliography is updated. Diagrams, figures and photos are all high quality and self explainig. The presence of Hepadnavirus in cats is a recent finding dated back only to 2018, little is still known about the life cycle, pathogenesis, epidemiology, etc. of this virus, so in my opinion every single case is worthy of being taken into consideration, even more if complete like this. It's a pity that it wasn't possible to perform a necropsy after the subject's death also to collect other samples.

Reply to Reviewer 2: We are extremely grateful to the referee for the positive feedback.

Round 2

Reviewer 1 Report

Comments and Suggestions for Authors

This is an interesting paper that has been improved with the revisions.